# Changes in Glial Support of the Hippocampus during the Development of an Alzheimer’s Disease-like Pathology and Their Correction by Mitochondria-Targeted Antioxidant SkQ1

**DOI:** 10.3390/ijms23031134

**Published:** 2022-01-20

**Authors:** Ekaterina A. Rudnitskaya, Alena O. Burnyasheva, Tatiana A. Kozlova, Daniil A. Peunov, Nataliya G. Kolosova, Natalia A. Stefanova

**Affiliations:** 1Institute of Cytology and Genetics, Siberian Branch of Russian Academy of Sciences (ICG SB RAS), 10 Lavrentyeva Ave., 630090 Novosibirsk, Russia; burnyasheva@bionet.nsc.ru (A.O.B.); kozlova@bionet.nsc.ru (T.A.K.); PeunovDA@bionet.nsc.ru (D.A.P.); kolosova@bionet.nsc.ru (N.G.K.); stefanovan@bionet.nsc.ru (N.A.S.); 2Faculty of Natural Sciences, Novosibirsk State University, 2 Pirogova St., 630090 Novosibirsk, Russia

**Keywords:** Alzheimer’s disease, astrocyte, microglia, mitochondria-targeted antioxidant, OXYS rats, SkQ1

## Abstract

Astrocytes and microglia are the first cells to react to neurodegeneration, e.g., in Alzheimer’s disease (AD); however, the data on changes in glial support during the most common (sporadic) type of the disease are sparse. Using senescence-accelerated OXYS rats, which simulate key characteristics of sporadic AD, and Wistar rats (parental normal strain, control), we investigated hippocampal neurogenesis and glial changes during AD-like pathology. Using immunohistochemistry, we showed that the early stage of the pathology is accompanied by a lower intensity of neurogenesis and decreased astrocyte density in the dentate gyrus. The progressive stage is concurrent with reactive astrogliosis and microglia activation, as confirmed by increased cell densities and by the acquisition of cell-specific gene expression profiles, according to transcriptome sequencing data. Besides, here, we continued to analyze the anti-AD effects of prolonged supplementation with mitochondria-targeted antioxidant SkQ1. The antioxidant did not affect neurogenesis, partly normalized the gene expression profile of astrocytes and microglia, and shifted the resting/activated microglia ratio toward a decrease in the activated-cell density. In summary, both astrocytes and microglia are more vulnerable to AD-associated neurodegeneration in the CA3 area than in other hippocampal areas; SkQ1 had an anti-inflammatory effect and is a promising modality for AD prevention and treatment.

## 1. Introduction

The hippocampus is one of the most vulnerable brain regions during the development of Alzheimer’s disease (AD), which is the most common type of dementia in the elderly worldwide [1]. Progressive neuronal death and decreased adult neurogenesis in the AD-affected hippocampus impair neuronal plasticity, which is crucial for the acquisition of certain types of contextual memory [2,3]. Astrocytes, which are the most common glial cells, and microglia—brain-resident macrophages—play a key part in neuronal survival and functioning [4] and are the first to react to stress in the central nervous system (CNS) [5], e.g., during the development of AD. In general, glial cells react to neurodegeneration in two ways: the first one includes activation (a reactive state for astrocytes and a proinflammatory state for microglia), and the second way is glial atrophy and a decline in glial activity [6]. Reactive astrogliosis is a complex condition with a gradual transition from normally functioning to significantly changed astrocytes; this condition has both general and disease-specific cellular and molecular characteristics [7,8]. The astrocyte response includes alterations in the gene expression profile, phagocytosis of synaptic and cellular debris, a recovery of the blood–brain barrier (BBB), and the acquisition of a reactive phenotype [9]. Moreover, at an advanced age, astrocytes become senescent: this means that they not only become proliferatively inactive but also undergo persistent changes in gene expression, such as the upregulation of proinflammatory factors (interleukins, chemokines, and matrix metalloproteinases) and growth factors (epidermal growth factor, basic fibroblast growth factor, and vascular endothelial growth factor) [10]. In this context, astrocytes also undergo alterations of epigenetic regulation and of mitochondrial homeostasis and show a disruption of energy and cellular metabolism [10]. Furthermore, senescent astrocytes from an aged brain are characterized by a higher production of reactive oxygen species [11], and hippocampal astrocytes undergo the most significant changes during AD [12]. The microglial response comprises their activation, consisting of morphological changes and the secretion of cytokines, chemokines, and reactive oxygen species; this process intensifies phagocytosis and increases BBB permeability and the recruitment of immune cells from the blood stream. Once activated, microglia lose some of their homeostatic functions and gain new functions instead [13]. Aged microglia have a primed phenotype characterized by an exaggerated and uncontrolled inflammatory response to an immune stimulus [14]. It is believed that the earliest detectable phase of microglial activation occurs at the preclinical stage of AD; as an indicator, microglial activation is concurrent with other inflammatory markers and astroglial activation indicators at later stages [15]. Microglia are broadly involved in the initiation, development, and progression of AD [16]. In particular, the phagocytic function of microglia is altered in AD [17], followed by decreased phagocytosis of amyloid-β (Aβ) and the formation of tau aggregates; simultaneously, microglia become activated and acquire a proinflammatory phenotype.

AD-associated changes in astrocytes and microglia have been extensively studied in transgenic mouse models of the familial type of AD, as well as in postmortem human brains; however, glial changes during the development of the most common—sporadic—type of AD (~95% of AD cases [18]) have been poorly examined so far [19]. In sporadic AD, amyloid deposition is likely to be much slower than that in familial AD and in its transgenic models. Additionally, it is possible that spatial and temporal properties of the glial response are different between human sporadic AD and human familial AD, i.e., are unique for each type of the disease [20]. Owing to the impact of systemic comorbidities, of the associated systemic inflammation, and of aging as a major risk factor of sporadic AD, a comprehensive study in suitable models is needed [21].

Recently, in a series of reports [22,23,24], we demonstrated the features of neurogenesis and neurotrophic and glial support of the brain in senescence-accelerated OXYS rats. OXYS rats were derived from the Wistar rat strain (normal healthy rats, control) and are characterized by accelerated senescence. Besides, OXYS rats spontaneously develop all the major signs of AD—Aβ accumulation, hyperphosphorylation of the tau protein, neuronal death, damage to synapses, and mitochondrial dysfunction—and largely reproduce known stages of the disease [25,26]. An important and distinct feature of OXYS rats is an overt neuronal loss in the hippocampus at the progressive stage of the AD-like pathology [25]. These neurodegenerative changes in the hippocampus of OXYS rats are accompanied by changes in the extracellular microenvironment of the neurogenic niche rather than by significant direct changes in the formation of new cells in the dentate gyrus (DG) [27]. Recently, we demonstrated that the density of amplifying neural progenitors (ANPs), which give rise to the neuronal cell lineage, is higher in OXYS rats than in Wistar rats during the completion of brain development, and, then, ANP density decreases [22]. Moreover, we showed the altered development of the hippocampus and prefrontal cortex in OXYS rats in an early postnatal period: a disturbance of astroglial support, a microglial deficiency, and a higher intensity of apoptosis during a period critical for the formation of a network among these brain structures [28]. We hypothesized that alterations in glial support observed early in life may contribute to some changes in glial density, structure, and function later in life during the development and progression of AD signs. The characteristics of astrocytes and microglia in the hippocampus of OXYS rats have not been researched so far.

In the present study, we investigated the hippocampal neurogenesis and glial support during the development and progression of AD signs in OXYS rats by means of previously obtained transcriptome sequencing data [29]; here, we focused upon glia-specific gene expression alterations. Using an immunohistochemical approach, we examined the density of neuronal progenitor cells in the DG and astroglial- and microglial-cell density in CA1 and CA3 regions of the hippocampus and in the DG. We chose the DG-CA3-CA1 regions because they form a tri-synaptic pathway starting from the entorhinal cortex and are highly important for memory consolidation. Furthermore, the DG is vulnerable to various stressors, partly because of the neurogenesis taking place in this brain region. The CA1 region is the most vulnerable brain structure during AD [30]. In addition, here, we continued to analyze the effect of prolonged supplementation (from age 12 to 18 months) with mitochondria-targeted antioxidant SkQ1 (plastoquinonyl-decyltriphenyl phosphonium) on neurogenesis and glial support in the hippocampus of OXYS rats. Previously, we have found that the anti-AD effects of SkQ1 are associated with an improvement in the activity of many signaling pathways and intracellular processes, including mitochondrial function [31,32]. Thus, SkQ1 may be a promising modality for the prevention and treatment of AD.

## 2. Results and Discussion

### 2.1. Age-Dependent Changes in the Density of Neuronal Progenitor Cells in the DG of OXYS and Wistar Rats; Effects of SkQ1

We showed that the ANP density is more than twofold lower in the DG of OXYS rats at 3 months of age (analysis of variance (ANOVA): F_1,12_ = 7.4, *p* < 0.02), indicating a lower intensity of neurogenesis at this age (Figure 1A). At the same time, the density of quiescent neural progenitors (QNPs), which may develop into a neuronal or glial cell lineage, remained unchanged (Figure 1B). Nonetheless, factorial ANOVA showed a significant main effect of the strain (genotype) on the QNP density in the DG: the parameter was higher in OXYS rats than in Wistar rats (F_1,28_ = 7.9, *p* < 0.009). The QNP density decreased with age in both rat strains (two-way ANOVA, main effect (age): F_1,28_ = 82.9, *p* < 0.0001) but was higher in 18-month-old OXYS rats than in 18-month-old Wistar rats (ANOVA: F_1,16_ = 28.5, *p* < 0.0001), thereby possibly reflecting the lower intensity of neurogenesis throughout the OXYS lifespan (Figure 1B). The ANP density was also influenced by the genotype (two-way ANOVA: F_1,28_ = 7.4, *p* < 0.01) and was lower in OXYS rats than in Wistar rats; the parameter naturally decreased with age in both rat strains (two-way ANOVA, main effect (age): F_1,28_ = 36.8, *p* < 0.0001), indicating an age-related slowdown of hippocampal neurogenesis. The rates of density changes were different between the strains (Figure 1A): from 3 to 18 months of age, the parameter decreased 10-fold in Wistar rats (ANOVA: F_1,13_ = 21.9, *p* < 0.0005) but only threefold in OXYS rats (ANOVA: F_1,15_ = 17.0, *p* < 0.001). This observation may support the hypothesis of the lower rate of neurogenesis in the DG of OXYS rats throughout their lifespan: the ANP density at 3 months of age was already lower in OXYS rats than in Wistar rats, and the age-related decrease was not as prominent in OXYS rats as in Wistar rats. Further research is needed to test this hypothesis and to clarify the possible connection between neurodegeneration and the rate of neurogenesis.

Previously, we have shown that, in OXYS rats, the development and progression of the AD-like pathology proceed concurrently with changes in the expression of hippocampal genes affecting all parameters of neurogenesis: cell proliferation, migration, incorporation into a synaptic network, and apoptosis [32]. Undoubtedly, neurogenesis is a highly energy-consuming process that requires healthy mitochondria. Mitochondrial abnormalities, together with synaptic degeneration, are the earliest and most prominent features of vulnerable neurons in the brain of AD patients [33,34,35]. For a decade, we have researched the role of mitochondria in the AD-affected brain using mitochondria-targeted antioxidant SkQ1 [31,32,36]. We have found that prophylactic and therapeutic effects of this antioxidant in all cases are associated with an improvement of the mitochondrial apparatus. Here, we did not find significant main effects of treatment with SkQ1 on the density of QNPs (F_1,31_ = 0.6, *p* = 0.44) and ANPs (F_1,31_ = 0.06, *p* = 0.81) in the DG of rats from both strains (Figure 1A,B). It should be noted that the results presented here and previously [31,32] were obtained in the same groups of animals for a more complete understanding of the anti-AD mechanisms of action of SkQ1 (all figures show only our new data). Thus, we can draw a conclusion that treatment with SkQ1 did not affect the density of neuronal progenitors but possibly facilitated the activation of the remaining undamaged neurons and synapses. This notion is supported by SkQ1-induced changes in the expression of OXYS hippocampal genes related to neuronal and synaptic processes [31]. On the other hand, it is well known that proper neuronal and synaptic function is supported by glial cells, including astrocytes and microglia [37,38]; besides, these cells are reported to play a crucial part in AD pathogenesis [39].

Figure 1C,D show representative immunohistochemical images of QNPs and ANPs in the DG of OXYS and Wistar rats.

### 2.2. Age-Dependent Changes in the Density of Astrocytes in the Hippocampus of OXYS and Wistar Rats; Effects of SkQ1

Here, for the first time, we estimated the density of astrocytes in the whole hippocampus of Wistar and OXYS rats at different stages of the progression of AD signs (at 3 and 18 months of age); we also investigated the effects of SkQ1 treatment from 12 to 18 months of age on the astrocyte density.

Factorial ANOVA revealed that the density of GFAP^+^ astrocytes was affected by age (main effect (age): F_1,46_ = 4.9, *p* < 0.03) and was not affected by the genotype (main effect (genotype): F_1,46_ = 2.7, *p* = 0.11), but there was an interaction between these factors (interaction effect: F_1,46_ = 7.8, *p* < 0.008). Indeed, we observed that, in Wistar rats, the astrocyte density decreased from 3 to 18 months of age (ANOVA: F_1,22_ = 12.3, *p* < 0.002), whereas only a slight increase in this parameter occurred in OXYS rats (Figure 2A). Pairwise comparisons showed that the density of astrocytes was significantly higher in OXYS rats than in Wistar rats at 18 months of age (Tukey’s test: *p* < 0.005).

An aging-related decrease in astrocyte density in CA1 and CA3 hippocampal areas in rodents has been documented elsewhere [40,41]. Our analysis of the astrocyte distribution in hippocampal regions of OXYS and Wistar rats revealed that the astrocyte density in the DG and CA1 area was influenced by age (two-way ANOVA, main effect (age): F_1_,_46_ = 4.4, *p* < 0.04 for DG; F_1,46_ = 12.6, *p* < 0.0009 for CA1) and by an interaction between the genotype and age (two-way ANOVA, interaction effect: F_1,46_ = 13.7, *p* < 0.0006 for the DG; F_1,46_ = 10.1, *p* < 0.003 for CA1), just as in the whole hippocampus (Figure 2C, upper panel; Appendix A). This was not the case for the CA3 region, where the density of astrocytes was higher in OXYS rats than in Wistar rats (two-way ANOVA, main effect (genotype): F_1,44_ = 6.0, *p* < 0.02) and was affected by age (two-way ANOVA, main effect (age): F_1,44_ = 6.9, *p* < 0.01). Regarding the DG, at 3 months of age, the density of astrocytes was lower in OXYS rats than in Wistar rats (Figure 2C; Tukey’s test: *p* < 0.05), mainly because of a decreased astrocyte density in granular and molecular layers (ANOVA: F_1,16_ = 5.7, *p* < 0.03 for the granular layer; F_1,16_ = 7.2, *p* < 0.02 for the molecular layer); however, at 18 months of age, the astrocyte density in the DG was higher in OXYS rats than Wistar rats (Tukey’s test: *p* < 0.004). Our data in some ways are consistent with the findings of Olabarria and colleagues [42]: those authors failed to detect changes in astroglial density in the hippocampus of 3xTG mice; however, they postulated a morphological atrophy of astrocytes already at an early stage of the familial-AD-like pathology—namely, in the DG—prior to Aβ plaque aggregation. We can hypothesize that the decline of astroglial density observed here contributes to the abnormally low intensity of neurogenesis seen at 3 months of age in the DG of OXYS rats.

In the CA1 region, the density of astrocytes decreased with age only in Wistar rats (Tukey’s test: *p* < 0.0008) because of a decrease in the pyramidal (Tukey’s test: *p* < 0.02) and molecular (Tukey’s test: *p* < 0.0005) layers, resulting in a higher astrocyte density at 18 months of age in OXYS rats than in Wistar rats (Figure 2C; Tukey’s test: *p* < 0.002). As for the CA3 area (Figure 2C), at 18 months of age, the density of astrocytes was higher in OXYS rats compared to Wistar rats, mostly because of a more than twofold higher astrocyte density in the pyramidal layer (Tukey’s test: *p* < 0.0002) and an increase in this parameter in the molecular layer (Tukey’s test: *p* < 0.003) of the CA3 region. Thus, in Wistar rats, the density of GFAP^+^ astrocytes naturally decreased with age in CA1 and CA3 regions; this was not the case in OXYS rats, where the astrocyte density in the DG even increased by 18 months of age. The unchanged astrocyte density in the hippocampus of OXYS rats at the advanced age, as compared to young OXYS rats, may be a consequence of a slower death of astrocytes or their re-entry into the cell cycle. Taking into account the increased cell death in the hippocampus of OXYS rats between 12 and 18 months of age [25,27], we can theorize that the latter scenario—namely cell cycle re-entry—takes place in OXYS rats; this hypothesis needs to be tested in future studies.

Treatment with SkQ1 caused an increase in the GFAP^+^ astrocyte density in Wistar rats (ANOVA: F_1,21_ = 6.6, *p* < 0.02) and did not affect it in OXYS rats (Figure 2A). The regional analysis showed that, in CA1, the astrocyte density was influenced by the genotype (two-way ANOVA, main effect (genotype): F_1,45_ = 10.4, *p* < 0.003) and treatment (two-way ANOVA, main effect (treatment): F_1,45_ = 4.1, *p* < 0.05). Additionally, we observed an interaction between the genotype and treatment, and this interaction affected the astrocyte density in the CA3 region of the hippocampus (two-way ANOVA, interaction effect: F_1,45_ = 8.6, *p* < 0.005; Figure 2D, upper panel; Appendix A). Indeed, the density of astrocytes in the CA3 region was significantly higher in OXYS rats than in Wistar rats (ANOVA: F_1,30_ = 23.6, *p* < 0.0001), and SkQ1 consumption decreased this parameter (ANOVA: F_1,24_ = 7.4, *p* < 0.02). At the same time, SkQ1 treatment led to an increase in the astrocyte density in the CA3 hippocampal area of Wistar rats (ANOVA: F_1,21_ = 7.1, *p* < 0.02). SkQ1 did not influence the astrocyte density in the DG of both rat strains. As a result of the crucial role of astrocytes in neurogenesis, this absence of an SkQ1 influence on their density may result in the absence of SkQ1 effects on neurogenesis, as discussed above.

Next, we counted vimentin-containing astrocytes. Vimentin is known to be expressed in astrocyte progenitors [43] and reactive astrocytes [44]. Although in a developing brain and in the DG, vimentin-positive astrocytes are thought to be astrocyte progenitors [43], in other hippocampal areas of the aged brain, these cells should be regarded as reactive [45]. Firstly, we found that, in the hippocampus, the density of vimentin-containing astrocytes is only much lower than the density of GFAP^+^ astrocytes (the *t* test for dependent samples: *p* < 0.0001). Then, we revealed that the density of vimentin^+^ astrocytes was not affected by age (two-way ANOVA: F_1,46_ = 1.3, *p* = 0.26) or the genotype (two-way ANOVA: F_1,46_ = 3.2, *p* = 0.08); however, there was an interaction between these factors (two-way ANOVA, interaction effect: F_1,46_ = 5.7, *p* < 0.02). Pairwise comparisons indicated that, at 18 months of age, the density of vimentin-containing astrocytes was more than 1.5-fold higher in OXYS rats relative to Wistar rats (Tukey’s test: *p* < 0.008; Figure 2B).

The regional analysis revealed that, in the CA1 area, the density of vimentin^+^ astrocytes increased with age (two-way ANOVA, main effect (age): F_1,46_ = 11.8, *p* < 0.001), and there was an interaction between the age and genotype (two-way ANOVA, interaction effect: F_1,46_ = 4.6, *p* < 0.04). In the CA3 region, the parameter also increased with age (two-way ANOVA, main effect (age): F_1,44_ = 13.4, *p* < 0.0007) and was higher in OXYS rats than in Wistar rats (two-way ANOVA, main effect (genotype): F_1,44_ = 5.9, *p* < 0.02). Pairwise comparisons showed that the density of vimentin-containing astrocytes in both CA1 and CA3 areas increased more than twofold from 3 to 18 months only in OXYS rats (Tukey’s test: *p* < 0.006 for both regions), becoming higher relative to Wistar rats (Tukey’s test: *p* < 0.02 for the CA1 region; *p* < 0.003 for the CA3 region). In both regions, the observed increase in this parameter was a consequence of a higher vimentin^+^ astrocyte density in the molecular layer (Figure 2C, lower panel; Appendix A).

Therefore, here, we revealed that, at the early stage of the spontaneously developing AD-like pathology, astrocyte deficiency rather than reactivity takes place in the hippocampus; by contrast, the progressive stage of this pathology is characterized by a higher density of GFAP^+^ astrocytes and vimentin^+^ astrocytes. Studies on the APPswePS1dE9 mouse model of AD have revealed that the cortical expression of GFAP and vimentin is increased during this pathology; however, other components of the intermediate-filament network, such as synemin and nestin, are not upregulated [46].

Concerning the effects of SkQ1, the density of vimentin^+^ astrocytes in the hippocampus increased only in Wistar rats after SkQ1 treatment (ANOVA: F_1,21_ = 6.0, *p* < 0.03; Figure 2B). Again, the CA1 area was the most affected region (Figure 2D, lower panel; Appendix A): the density of vimentin-containing astrocytes was higher in OXYS rats than in Wistar rats (two-way ANOVA, main effect (genotype): F_1,45_ = 6.3, *p* < 0.02) and increased after SkQ1 supplementation (two-way ANOVA, main effect (treatment): F_1,45_ = 8.6, *p* < 0.005). In the CA3 region, the SkQ1 treatment exerted opposite effects on the density of vimentin^+^ astrocytes (Figure 2D, lower panel; Appendix A): treatment with SkQ1 decreased this parameter in OXYS rats (ANOVA: F_1,24_ = 5.1, *p* < 0.04) but increased it in Wistar rats (ANOVA: F_1,21_ = 10.1, *p* < 0.005). Representative immunohistochemical images of GFAP^+^ and vimentin^+^ astrocytes in the hippocampus of OXYS and Wistar rats are provided in Figure 2E,F.

Taken together, the results suggested that, at the progressive stage of the AD-like pathology, OXYS rats manifest the signs of astrogliosis; the most affected region of the hippocampus is CA3. Treatment with SkQ1 restored the astrocyte density in the CA3 region of OXYS rats (almost to the level seen in the normal control strain, Wistar rats), thus probably alleviating astrogliosis. On the other hand, the astrocyte density increased after SkQ1 treatment in the hippocampus of Wistar rats.

### 2.3. Genes Differentially Expressed between OXYS and Wistar Rats in Astrocytes

We analyzed the hippocampal transcriptome using lists of cell-type specific genes created on the basis of data from single-cell high-throughput RNA sequencing (single-cell RNA-Seq) [47].

According to the RNA-Seq data, there were 10 differentially expressed genes (DEGs) specific for astrocytes in the hippocampus of OXYS rats compared to Wistar rats at 5 months of age (Appendix A). Among them, four genes had a higher expression (*Fosb*, *Itga7*, *Slc1a4*, and *Tom1l1*), and six genes had a lower expression in OXYS rats (*Cth*, *Cyp1b1*, *Evc*, *Gem*, *Itih3*, and *Ptplb*). It is important to point out that four genes (*Evc*, *Gem*, *Itga7*, and *Slc1a4*) are known to be associated with the cell membrane; the proteins encoded by these genes take part in actin cytoskeleton organization and interactions with the extracellular matrix. Of note, *Itih3* codes for a component of the extracellular matrix. Proteins encoded by genes *Cth*, *Cyp1b1*, and *Ptplb* are enzymes that participate in the metabolism of amino acids and fatty acids. We can conclude that interactions with the extracellular matrix and some metabolic functions may become abnormal in the astrocytes of OXYS rats during the manifestation of AD signs (Figure 3A).

During the progression of the AD-like pathology from 5 to 18 months of age, we found 218 DEGs specific for astrocytes in OXYS rats and 167 DEGs in Wistar rats at 18 months of age as compared to an age of 5 months (Appendix A); among them, 136 DEGs were common between the two strains and had the same pattern of altered expression. Forty-eight DEGs found only in OXYS rats decreased their expression, whereas 34 DEGs increased their expression from 5 to 18 months of age. We revealed that, among the 82 time-dependent DEGs seen only in OXYS rats, 36 genes encode membrane proteins, and products of 12 genes are associated with the extracellular space. We noticed that the expression of genes *Rb1* and *Dpf3* decreased, whereas the expression of *Sept9* increased with age. Due to the fact that the products of *Rb1* and *Dpf3* are negative regulators of the cell cycle—whereas the protein encoded by *Sept9* is involved in cytokinesis and cell cycle control—we can theorize that the cell cycle control in astrocytes weakened during the progression of AD signs in OXYS rats (Figure 3B). It is widely accepted that, during AD, reactive astrocytes change their intermediate-filament network but do not proliferate [48]. On the other hand, the weakness of the cell cycle control may be caused by Aβ, as demonstrated in neurons [49], and drives aberrant cell cycle re-entry and DNA duplication without cytokinesis rather than proper cell division. Furthermore, the observed increase in the vimentin^+^ cell density, together with an elevated *Gfap* expression and *Kcnj10* underexpression, may point to the development of reactive astrogliosis from 5 to 18 months of age in OXYS rats; this is because such expression alterations are among potential markers of reactive astrocytes [50]. Additionally, we documented the downregulation of *Megf10* and *Aqp4* in astrocytes by 18 months of age. These genes code for proteins crucial for Aβ phagocytosis and Aβ removal throughout the BBB in the blood stream and lymphatic vessels. Due to the fact that Aβ toxicity is considered to be the major damaging factor during AD, the impairment of Aβ clearance, along with its overproduction, may be a cause of AD [51]. The key role in Aβ clearance is played by astrocytes, through which, Aβ is transferred to the blood stream and lymphatic drainage vessels [52]. Thus, the downregulation of *Aqp4* and *Megf10* in hippocampal astrocytes by 18 months of age may result in an altered Aβ clearance and may contribute to the Aβ pathology observed in OXYS rats [25].

At 18 months of age, there were 45 DEGs specific for astrocytes in OXYS rats relative to Wistar rats (Appendix A); among them, 18 genes had a higher expression, and 27 genes had a lower expression in OXYS rats. Among the DEGs, five genes (*Agt*, *Cbs*, *Gfap*, *Sept9*, and *Slc25a34*) turned out to be upregulated because of their age-related increase only in OXYS rats, and the *Chsy1* gene had a higher expression because of the absence of its age-related decrease (this decrease is characteristic for Wistar rats). Besides, the expression of seven genes (*Alpk1*, *Igsf11*, *Megf10*, *Pdgfrb*, *Rgs20*, *Slc30a10*, and *Tmc7*) was lower as a consequence of an age-related decrease only in OXYS rats; the expression of *Dclk2* was lower in OXYS rats because of the absence of its age-related upregulation seen in Wistar rats. Moreover, there were seven astrocyte-specific DEGs common between 5 and 18 months of age in OXYS rats: five of them were found to be downregulated (*Cth*, *Cyp1b1*, *Evc*, *Gem*, and *Itih3*), and two genes were upregulated (*Fosb* and *Tom1l1*). Similar to the age of 5 months, at 18 months of age, nearly half of the DEGs were found to encode membrane proteins (20 of 45 genes), and 10 genes proved to encode proteins localized in the extracellular space. Furthermore, the expression of genes coding for cytoskeletal proteins in astroglia (*Gfap*, *Myo6*, *Pls1*, and *Sept9*) was found to change in OXYS rats relative to Wistar rats. Taken together, these results may point to an altered structure of astrocytes and their interaction with extracellular molecules (Figure 3B). A higher expression of genes *Fox*, *Foxb*, and *Sept9* and a lower expression of *Megf10* and *Zfp217* in OXYS rats relative to Wistar rats again indicate the dysregulation of cell cycle re-entry. Astrocytes are known to induce the formation of excitatory synapses by secreting some protein factors, one of which being SPARCL1 [53]. Additionally, the MEGF10 protein serves as a phagocytic receptor mediating synaptic phagocytosis [54]. OXYS rats featured a lower expression of *Sparcl1* and *Megf10* in the hippocampus at 18 months of age (as compared to Wistar rats). This aberration may contribute to altered synaptic plasticity—which means a decreased synaptic formation and pruning—at the progressive stage of the AD-like pathology, as demonstrated earlier [55]. Additionally, the *Rgs20* gene (encoding a regulator of G_q_ and G_i_ protein signaling) is downregulated in OXYS rats relative to Wistar rats. The activation of the G_q_ protein induces an increase in intracellular Ca^2+^ concentration [56]; thus, the underexpression of its regulator may cause a decrease in intracellular Ca^2+^ concentration. The observed change in gene expression may be regarded as a compensatory reaction designed to reduce the higher Ca^2+^ concentration in the astrocytes. In this regard, it is known that the intracellular Ca^2+^ level is elevated in reactive astrocytes located near amyloid plaques [57]. This assumption needs to be verified.

Insofar as astrocyte senescence is induced by Aβ and oxidative stress [58], the influence of SkQ1 supplementation on the astrocyte gene expression profile is to be expected. Indeed, previously, we have shown that major effects of SkQ1 on astrocytes are related to the regulation of processes such as biosynthesis, cell proliferation, transcription, apoptosis, tube development, and blood vessel development [31]. Here, we revealed that SkQ1 treatment prevented the age-associated (from 5 to 18 months of age) upregulation of four genes specific for astrocytes (*Hhipl1*, *Jun*, *Paqr7*, and *Sept9*) and downregulation of seven genes (*Tmem47*, *Alpk1*, *Megf10*, *Pdgfrb*, *Rgs20*, *Slc30a10*, and *Tmc7*) in OXYS rats. This means that we failed to detect differences in the expression of these genes between SkQ1-treated OXYS rats at 18 months of age and OXYS rats at 5 months of age (Appendix A). Moreover, supplementation with SkQ1 normalized the expression of five upregulated genes (*Chsy1*, *Hes1*, *Plagl1*, *Ppap2b*, and *Tpbg*) and 12 downregulated genes (*Egfr*, *Elovl5*, *Evc*, *Gem*, *Il33*, *Kcna2*, *Myo6*, *Pls1*, *Scara3*, *Slc15a2*, *Slc7a11*, and *Slc7a2*) in OXYS rats at 18 months of age. This means that we could not find any differences in the expression of these genes at 18 months of age between SkQ1-treated OXYS rats and untreated Wistar rats (healthy control rats). The majority (21 of 28) of genes whose expression was restored by SkQ1—which means the prevention of both age-related differences (from 5 to 18 months of age in OXYS rats) and interstrain differences (OXYS vs. Wistar rats at 18 months of age)—code for proteins from the extracellular space or plasma membrane. These data indicate that SkQ1 improved the astrocyte structure and interaction with extracellular stimuli (Figure 3B). In addition, SkQ1 restored the expression of almost all genes associated with the cytoskeleton (Figure 3B), with the exception of *Gfap*, in line with the absence of SkQ1’s influence on the GFAP^+^ cell density in OXYS rats. It is important to point out that SkQ1 treatment prevented abnormal *Megf10* underexpression, thereby possibly improving Aβ phagocytosis and the pruning of dysfunctional synapses [32].

### 2.4. Age-Dependent Changes in the Density of Microglia in the Hippocampus of OXYS and Wistar Rats; Effects of SkQ1

We measured the density of resting (Iba1^+^) and activated (Iba1^+^CD68^+^) microglia, as well as the total microglial density, in several hippocampal regions of OXYS and Wistar rats at 3 and 18 months of age. We found that the total microglial density in the whole hippocampus was affected only by the genotype (two-way ANOVA, main effect [genotype]: F_1,59_ = 6.3, *p* < 0.02) and was higher in OXYS rats than in Wistar rats. In the meantime, the density of resting microglia in the whole hippocampus was influenced by neither the genotype (two-way ANOVA: F_1,63_ = 1.6, *p* = 0.21) nor age (two-way ANOVA: F_1,63_ = 1.7, *p* = 0.19). By contrast, the density of activated microglia was affected by the genotype (two-way ANOVA, main effect (genotype): F_1,61_ = 4.3, *p* < 0.05) and was higher in OXYS rats than in Wistar rats (Figure 4A). Thus, we may conclude that the increased total microglial density in OXYS rats was due to a higher density of activated microglia.

As for the regional analysis (Figure 4B; Appendix A), we noticed a significant main effect of the hippocampal region (multifactorial ANOVA: F_1,59_ = 8.0, *p* < 0.007): in the CA3 area, the microglial density was the highest among all tested regions in both rat strains. In the DG, the total microglial density significantly increased from 3 to 18 months of age (two-way ANOVA, main effect (age): F_1,27_ = 15.9, *p* < 0.0005), largely because of a higher resting-microglia density (two-way ANOVA, main effect (age): F_1,27_ = 57.1, *p* < 0.0001). Neither the genotype nor age affected the total microglial density in the CA3 region (two-way ANOVA: F_1,27_ = 0.4, *p* = 0.55 for effects of the genotype; F_1,27_ = 1.3, *p* = 0.26 for effects of age). Nonetheless, there was an interaction between these factors in the CA1 area (two-way ANOVA, interaction effect: F_1,28_ = 4.7, *p* < 0.04). Indeed, this parameter rose with age only in OXYS rats (ANOVA: F_1,17_ = 13.5, *p* < 0.002) via an increase in the resting-microglia density (ANOVA: F_1,17_ = 107.1, *p* < 0.0001), whereas, in Wistar rats, these parameters remained unchanged.

It is widely accepted that activated microglia secrete factors that induce the transformation of astrocytes to the reactive state; reactive astrocytes, in turn, intensify the acquisition of the activated phenotype by microglia [50]. Next, we analyzed a possible correlation between the densities of vimentin^+^ astrocytes (considered “reactive”) and CD68^+^ microglia (regarded as “activated”) in hippocampal regions of OXYS and Wistar rats at an advanced age (meaning 18 months; Appendix A). As expected, we detected a positive correlation between the parameters in the CA1 area of Wistar rats (correlation: r = 0.84, *p* < 0.04). Meanwhile, a strong negative correlation between the reactive-astrocyte density and activated-microglia density was detectable in the CA3 region of OXYS rats (correlation: r = −0.96, *p* < 0.002). This result may indicate the prevalence of only one type of cell with a proinflammatory phenotype in the CA3 area during the progressive stage of the AD-like pathology in OXYS rats, likely owing to the recruitment of cells of the same type. This observation requires further investigation.

Thus, here, we found an increased density of CD68^+^ activated microglia during both the manifestation and progression of the AD-like pathology. These data suggest that microglial changes occur before Aβ accumulation does, and progress during the disease-like pathology. Earlier, we detected the signs of destructive changes in hippocampal microglial cells, as well as in the astrocytes of 18-month-old OXYS rats: a low electron density of the hyaloplasm, enlarged mitochondria with partial or full disintegration of cristae, numerous vacuoles, membrane complexes, and lipofuscin inclusions [55]. The alterations in the structure of microglial cells reflect the loss of their regulatory functions related to migration, the clearance of cellular waste, and ultimately, neuronal survival.

Next, we assayed the effects of SkQ1 treatment on microglial density. Factorial ANOVA detected only insignificant effects of the genotype and treatment on the total microglial density (main effect (genotype): F_1,28_ = 3.3, *p* = 0.079; main effect (treatment): F_1,28_ = 3.9, *p* = 0.059, respectively). Even though the total microglial density remained unchanged, SkQ1 supplementation affected the resting/activated microglia ratio (Figure 4A; Appendix A). In particular, SkQ1 treatment raised the resting-microglia density (two-way ANOVA, main effect (treatment): F_1,28_ = 13.1, *p* < 0.001) and lowered that of activated microglia (two-way ANOVA, main effect (treatment): F_1,28_ = 8.8, *p* < 0.006), thereby exerting an anti-inflammatory effect. Pairwise comparisons revealed that SkQ1 increased the density of resting microglia in the whole hippocampus of both rat strains (Tukey’s test: *p* < 0.02 for Wistar rats; *p* < 0.002 for OXYS rats) and reduced the activated-microglia density only in OXYS rats (Tukey’s test: *p* < 0.04).

Supplementation with SkQ1 eliminated the correlations between the reactive-astrocyte density and activated-microglia density (see above) in hippocampal regions of both rat strains (Appendix A).

After that, we examined the density of pyknotic nuclei and their percentage phagocyted by microglia (Appendix A; Appendix A). We revealed that neither the genotype nor treatment affected the percentage of pyknotic nuclei phagocyted by microglia. On the contrary, treatment with SkQ1 lowered the total density of pyknotic nuclei in the hippocampus (two-way ANOVA, main effect [treatment]: F_1,31_ = 7.5, *p* < 0.01). Thus, SkQ1 treatment had an anti-inflammatory effect in the hippocampus of rats from both strains, as evidenced by a decreased activated-microglia density and increased resting-microglia density, partly because of SkQ1’s contribution to the decrease in the pyknotic-nuclei density.

### 2.5. Genes Differentially Expressed between OXYS and Wistar Rats in Microglia

Next, we analyzed the expression of genes specific to microglia in the hippocampus during the development and progression of AD signs in OXYS rats and evaluated the possibility of SkQ1’s influence on the expression of these genes. At 5 months of age, there were four DEGs specific to microglia in the hippocampus of OXYS rats compared to Wistar rats. The expression of *Map3k14* and *Aldh16a1* was higher, and the expression of *Cd48* and *Cyp4v3* was lower in OXYS rats. Given that the *Map3k14* gene encodes a kinase of NF-κB, its overexpression may enhance the activation of the proinflammatory TNF pathway; at the same time, *Cd48* underexpression may mean a weaker activation of microglia by extracellular stimuli. Therefore, the decrease in the *Cd48* mRNA level may be considered to be a response that compensates for the increased activation of the TNF pathway (Figure 5A).

As for age-related gene expression changes—from 5 to 18 months of age—we found 217 DEGs specific for microglia in Wistar rats and 185 DEGs in OXYS rats (Appendix A). Among the time-dependent DEGs in Wistar rats, 14 genes decreased their expression, whereas 203 genes increased their expression between these ages. In OXYS rats, 26 genes diminished and 159 genes increased their expression. From 5 to 18 months of age, 138 DEGs were common between OXYS and Wistar rats. In both rat strains, we observed an activation of the following pathways: glycan degradation (*HexA*, *Hexb*, and *Man2b1*) and glycosaminoglycan degradation (*Glans*, *Hexa*, *Hexb*, and *Naglu*; FDR < 0.01); lysosome formation and function (*Cd68*, *Ctsa*, *Ctsb*, *Ctsd*, *Ctsf*, *Ctss*, *Ctsz*, *Glans*, *Hexa*, and *Hexb*); and apoptosis (*Ctsb*, *Ctsd*, *Ctsf*, *Ctss*, *Ctszb*, *Ctszm*, *Gadd46b*, *Mcl1*, and *Nfkb1*). There were 47 genes that changed their expression from 5 to 18 months of age only in OXYS rats; among them, 12 genes decreased expression, and 35 genes were upregulated. Among these genes, 21 are associated with the cell membrane, and six genes with the cytoskeleton. Furthermore, an analysis of Gene Ontology categories (biological processes) revealed that 19 genes are related to cell communication.

At the progressive stage of neurodegeneration—meaning at 18 months of age—we found 33 strain-dependent DEGs specific to microglia in the hippocampus of OXYS rats relative to Wistar rats (Appendix A). Among them, 14 genes had a higher expression, and 19 genes had a lower expression in OXYS rats. Just as at the age of 5 months, OXYS rats at 18 months had a higher mRNA level of *Map3k14* and diminished mRNA levels of *Cd48* and *Cyp4v3* compared with Wistar rats. Aside from *Map3k14*, OXYS rats manifested an overexpression of *Cebp* and *Junb*; products of all these genes take part in the triggering of TNF signaling. Meanwhile, the expression of *Pik3cg*, *Spp1*, and *Cdk6* was lower in OXYS rats than in Wistar rats; proteins encoded by these genes are components of mTOR signaling. In addition, the expression of *Ubc* was lower in OXYS rats; this gene codes for a polyubiquitin precursor. The mRNA level of *Nhlrc3*—another gene whose product is involved in ubiquitination—was lower in OXYS rats than in Wistar rats. The ubiquitin proteasome system in microglia is crucial for Aβ degradation [59]. The upregulation of *Plekho1* and downregulation of *Skap2* mRNA expression are suggestive of greater actin stabilization, which, in turn, may worsen cellular and process motility, which are key features for the microglial response to AD [60]. The level of *Ctss* mRNA was significantly lower in OXYS rats than in Wistar rats; this gene encodes the cathepsin S protein, which belongs to the peptidase C1 family and may participate in the degradation of antigenic proteins for presentation to MHC class II molecules. To sum up, we found signs of an activation of proinflammatory signaling cascades and an altered interaction of microglia with the extracellular space during the progression of AD signs in OXYS rats (Figure 5B).

As for the effects of SkQ1 treatment, previously, we reported that the following pathways are influenced: a response to an organic substance, the regulation of cell death, GTPase regulator activity, and ion homeostasis [31]. Here, we demonstrated that treatment with SkQ1 prevented the age-associated increase (from 5 to 18 months of age) in the expression of seven genes (*Plekho1*, *Arhgap22*, *Dok3*, *Fam110a*, *Rogdi*, *Rtn4rl1*, and *Snta1*) and downregulation of four genes (*Hk2*, *Pik3cg*, *Slc11a1*, and *Ubc*) specific for microglia in OXYS rats: this means that we could not find differences in the expression of these genes between SkQ1-treated 18-month-old OXYS rats and 5-month-old OXYS rats (Appendix A). Furthermore, the expression of *Dusp1* increased with age (from 5 to 18 months of age) in SkQ1-treated OXYS rats, just as in Wistar rats. Additionally, SkQ1 abrogated the upregulation of five genes (*Cebpb*, *Egr2*, *Rgs14*, *Tmem206*, and *Tor3a*) in OXYS rats: this means that we failed to detect any differences in the expression of these genes at 18 months of age between SkQ1-treated OXYS rats and untreated Wistar rats (healthy control strain). Taken together, our results mean that the treatment with SkQ1 partly reversed the pathological alterations of mTOR signaling, of the ubiquitin proteasome system, and of actin cytoskeleton reorganization (Figure 5B).

Summarizing the findings, we can conclude that the development of the AD-like pathology in OXYS rats is accompanied by such signs as a lower intensity of hippocampal neurogenesis accompanied by an insufficient astrocyte density in the neurogenic niche. Another conclusion is that the progressive stage of the AD-like pathology in OXYS rats is characterized by astrogliosis and microglial activation in the hippocampus. According to our results, both astrocytes and microglia are more vulnerable to AD-associated neurodegeneration in the CA3 area than in other hippocampal areas. Possible functional consequences of the changes in glia density and altered gene expression profiles are being investigated further. The treatment with mitochondria-targeted antioxidant SkQ1 from 12 to 18 months of age decreased the signs of astrogliosis in the hippocampus of OXYS rats, but increased these signs in Wistar rats. Additionally, SkQ1 treatment had an anti-inflammatory effect in the hippocampus of rats from both strains; this phenomenon was mediated by the observed shift of the resting-/activated-microglia density ratio toward the predominance of resting cells. The presented effects of SkQ1 may underline its ability to prevent neuronal loss and synaptic damage, to enhance a neurotrophic supply, and to decrease Aβ protein levels and tau hyperphosphorylation in the hippocampus [32].

## 3. Materials and Methods

### 3.1. Animals

The OXYS rat strain was developed at the Institute of Cytology and Genetics (ICG), SB RAS (Novosibirsk, Russia), from a Wistar stock by selection for susceptibility to the cataractogenic effect of a galactose-rich diet and brother–sister mating of highly susceptible rats. Currently, we have the 117th generation of OXYS rats. These senescence-accelerated strain and age-matched male Wistar rats (parental normal strain) were obtained from the Breeding Experimental Animal Laboratory of the ICG SB RAS (Novosibirsk, Russia). The animals were kept under standard laboratory conditions (22 ± 2 °C, 60% relative humidity, and the 12 h light/12 h dark cycle) and had ad libitum access to standard rodent feed (PK-120-1, Laboratorsnab, Ltd., Moscow, Russia) and water. The animals were kept in groups of four per cage. The study was conducted according to Directive 2010/63/EU of the European Parliament and of the Council of 22 September 2010 and was approved by the Commission on Bioethics at the ICG SB RAS (# 34 of 15 June 2016), Novosibirsk, Russia.

### 3.2. SkQ1 Administration

We used only male rats to avoid the effects of the hormonal cycle characteristic of female rats. To assess the influence of oral SkQ1 administration on the neurogenesis in the DG and on glial support of the hippocampus in OXYS rats, we randomly assigned 12-month-old male OXYS and Wistar rats to one of two groups (six rats per group, 2 × 2 = 4 groups). One group consumed a control diet (dried bread slices in addition to the standard rodent feed), and the other consumed the same diet supplemented with a SkQ1 solution (250 nM SkQ1 per kg of body weight) daily from 12 to 18 months of age. Each rat received dried bread slices (with or without the SkQ1 solution in them) individually. We added a solution of SkQ1 onto a dried bread slice; the amount of SkQ1 was normalized to rat body weight. Then, each rat in the cage received its own bread slice and ate it whole. Adding the SkQ1 solution to a bread slice during an experiment is a simple and nonstressful way to administer an exact amount of SkQ1 per os.

### 3.3. Tissue Preparation and Immunohistochemistry

For assaying the age-related changes in the glial support of the hippocampus, OXYS and Wistar rats were euthanized by CO_2_ asphyxiation and decapitated at either 3 or 18 months of age; the animals that had been consuming SkQ1 were euthanized by CO_2_ asphyxiation and decapitated at 18 months of age. The brains were carefully excised, and the hemispheres were separated and immediately fixed in 4% paraformaldehyde in phosphate-buffered saline (PBS) at room temperature (RT) for 48 h, followed by cryoprotection in 30% sucrose in PBS at 4 °C for 48 h. Then, the brains were frozen and stored at −70 °C until further processing.

Brain sagittal sections (20 μm thick) of OXYS and Wistar rats (*n* = 4 to 6 per strain, age, and experimental group) were prepared on a Microm HM-505 N cryostat (Microm, Walldorf, Germany) at −20 °C and were transferred onto polysine-glass slides (Menzel-Glaser, Braunschweig, Germany). After serial washes with PBS, the slices were incubated at RT for 15 min in PBS-plus (PBS with 0.1% of Triton X-100) and for 1 h in 3% bovine serum albumin (BSA; cat. # A3294, Sigma-Aldrich, St. Louis, MO, USA) in PBS to permeabilize the tissues and to block nonspecific binding sites, and were then incubated overnight with primary antibodies at 4 °C. The primary antibodies were all diluted 1:250 with 3% BSA in PBS; these were antibodies to nestin, vimentin, glial fibrillary acid protein (GFAP), Iba1, and CD68 (cat. ## ab6142, ab24525, ab7260, ab5076, and ab31630, respectively, Abcam, Cambridge, MA, USA). After several washes with PBS, the slices were probed with secondary antibodies conjugated with Alexa Fluor 488 or 568 (cat. ## ab150073 and ab175472, respectively, Abcam) in PBS (1:250) for 1 h at RT and were then washed in PBS. The slices were coverslipped with the Fluoroshield mounting medium containing 4′,6-diamidino-2-phenylindole (DAPI; cat. # ab104139, Abcam). Negative controls were processed in an identical manner, except that a primary antibody was not included. The nestin, vimentin, GFAP, Iba1, and CD68 signals were detected under a microscope with a 40× objective lens (Axioskop 2 plus, Zeiss, Oberkochen, Germany). The microscopy was conducted at the Multi-Access Center for Microscopy of Biological Objects (ICG SB RAS, Novosibirsk, Russia). Identification of brain structures (CA1, CA3, and DG regions of the hippocampus) was performed according to Paxinos and Watson (Lateral 0.40 to Lateral 0.90 mm) [61]. Identification of cell types was carried out according to protein markers described by Encinas and colleagues [62].

Total numbers of QNPs (nestin^+^ vimentin^+^ cells), ANPs (nestin^+^ cells), GFAP^+^ and vimentin^+^ astrocytes, and resting (Iba1^+^) and activated (Iba1^+^CD68^+^) microglia were determined by means of the ZEN 3.1 software (Zeiss, Oberkochen, Germany). To evaluate the density of QNPs and ANPs, the total number of counted cells was divided by DG area and then averaged in each group of 2–3 tissue slices per animal and presented as the number of cells per 10,000 μm^2^. To assess the density of astrocytes and resting and activated microglia, the total number of counted cells was divided by the area of the hippocampus and then averaged in each group of 2–3 slices per animal and presented as the number of cells per 10,000 μm^2^. To determine the percentage of dying cells consumed by microglia, we counted all pyknotic nuclei and the pyknotic nuclei covered by microglial cytoplasm.

### 3.4. RNA-Seq Analysis

The RNA-Seq data were obtained earlier, as described elsewhere [29]. The lists of cell-specific DEGs were created on the basis of data on single-cell RNA-Seq [47]. Pathway analysis of the DEGs was conducted by means of the WEB-based Gene Set Analysis Toolkit [63] using KEGG pathways (https://www.genome.jp/kegg/ accessed on 18 November 2021).

### 3.5. Statistics

The data were subjected to two-way ANOVA in Statistica 8.0 (TIBCO Software Inc., Palo Alto, CA, USA). The genotype (strain) and age, as well as genotype and treatment, were chosen as independent factors. Tukey’s test was applied to significant main effects and interactions in order to assess differences between some sets of means. The *t* test for dependent samples was performed for a dependent-pair comparison within the same animal. Correlation analysis was used to estimate associations between parameters. The data are presented as the mean ± standard error of the mean (SEM). The differences were considered statistically significant at *p* < 0.05.

## Figures and Tables

**Figure 1 ijms-23-01134-f001:**
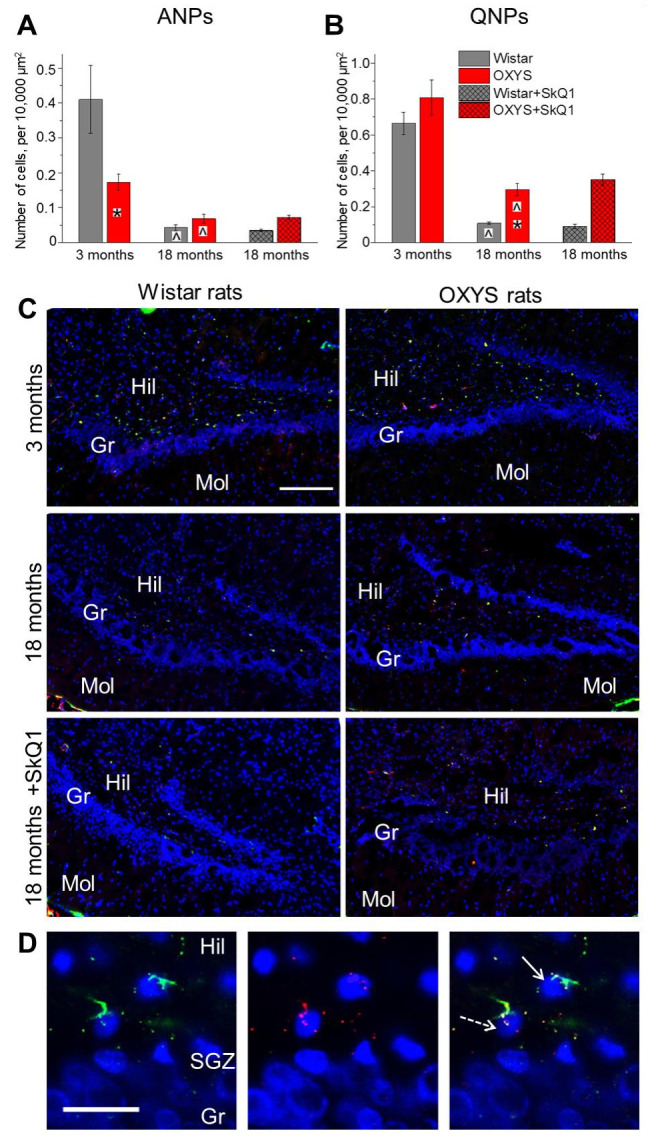
The density of neuronal progenitors in the dentate gyrus (DG) of OXYS and Wistar rats at 3 and 18 months of age, as well as after dietary supplementation with SkQ1 (a mitochondria-targeted antioxidant). (**A**) The density of amplifying neural progenitors (ANPs) was lower in OXYS rats at 3 months of age relative to Wistar rats. (**B**) OXYS rats had higher quiescent neural progenitor (QNP) density at 18 months of age than Wistar rats. (**C**) Immunohistochemical staining of the DG with antibodies against nestin (green) and vimentin (red). The scale bar is 200 µm. (**D**) A representative image of an ANP (arrow) stained for nestin (green) and of a QNP (dashed arrow) stained for nestin (green) and vimentin (red); the DG of the hippocampus of a Wistar rat at 3 months of age; the scale bar is 20 µm. 4′,6-Diamidino-2-phenylindole (DAPI, blue) indicates cell nuclei (**C**,**D**). Gr: granular layer; Hil: hilus; Mol: molecular layer; SGZ: subgranular zone. The data (**A**,**B**) are presented as mean ± SEM; * *p* < 0.05 for differences between the strains; ^ *p* < 0.05 for differences from a previous age.

**Figure 2 ijms-23-01134-f002:**
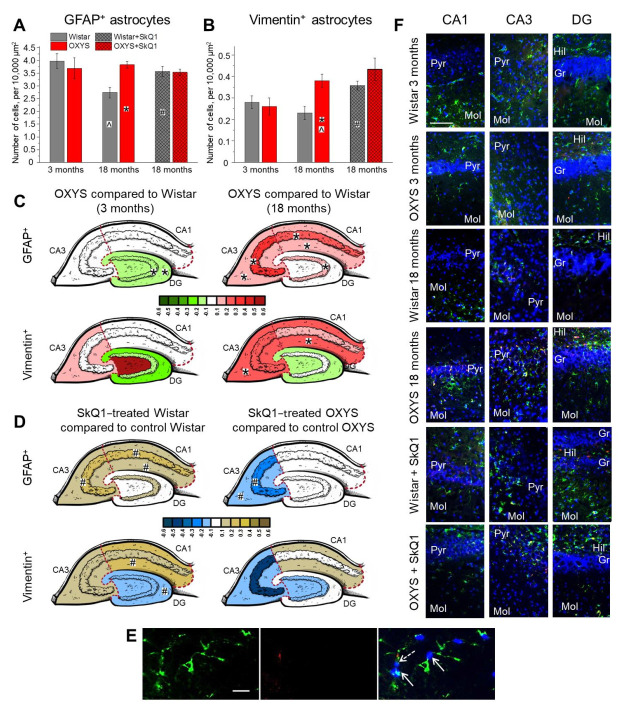
The density of GFAP^+^ (**A**) and vimentin^+^ (**B**) astrocytes in the hippocampus of OXYS and Wistar rats at 3 and 18 months of age, as well as after SkQ1 treatment. Changes in GFAP^+^ and vimentin^+^ cell density in the regions of hippocampus of OXYS rats compared to Wistar rats (**C**) and in SkQ1-treated compared to untreated animals (**D**) are presented schematically. GFAP^+^ and vimentin^+^ cell density in each region and layer is coded as a heatmap on an *lg* scale from −0.6 to 0.6. (**E**) A representative image of GFAP^+^ cells (green; indicated by the arrow) and vimentin^+^ cells (red; indicated by the dashed arrow) (the molecular layer of the CA1 area in the hippocampus of a Wistar rat at 3 months of age). The scale bar is 20 µm. (**F**) Images of GFAP^+^ cells (green) and vimentin^+^ cells (red) in the hippocampal regions of OXYS and Wistar rats at 3 and 18 months of age, as well as after SkQ1 supplementation; the scale bar is 100 µm. 4′,6-Diamidino-2-phenylindole (DAPI, blue) indicates cell nuclei (**E**,**F**). Gr: granular layer; Hil: hilus; Mol: molecular layer; Pyr: pyramidal layer. The data (**A**,**B**) are presented as mean ± SEM; * *p* < 0.05 for differences between the strains; ^ *p* < 0.05 for differences from a previous age; ^#^
*p* < 0.05 for effects of SkQ1 treatment.

**Figure 3 ijms-23-01134-f003:**
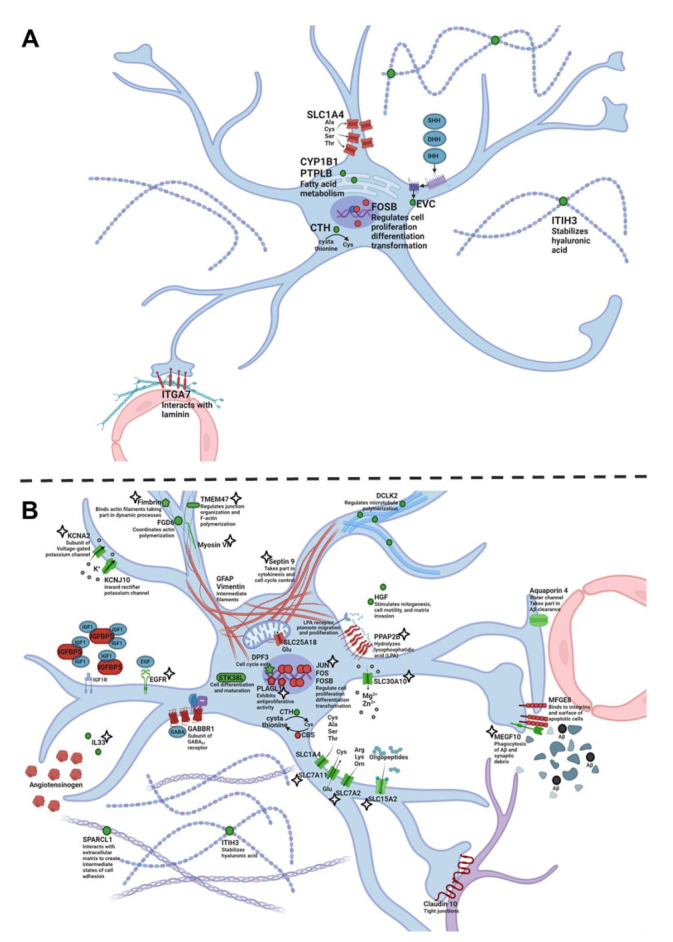
An astrocyte of 5-month-old (**A**) and 18-month-old (**B**) OXYS rats. Proteins with decreased gene expression—as compared to Wistar rats or to previous age—are highlighted in green; proteins with increased gene expression are red. Stars (**B**) indicate the genes whose expression was altered by SkQ1. The illustration was created on BioRender.com (accessed on 15 November 2021).

**Figure 4 ijms-23-01134-f004:**
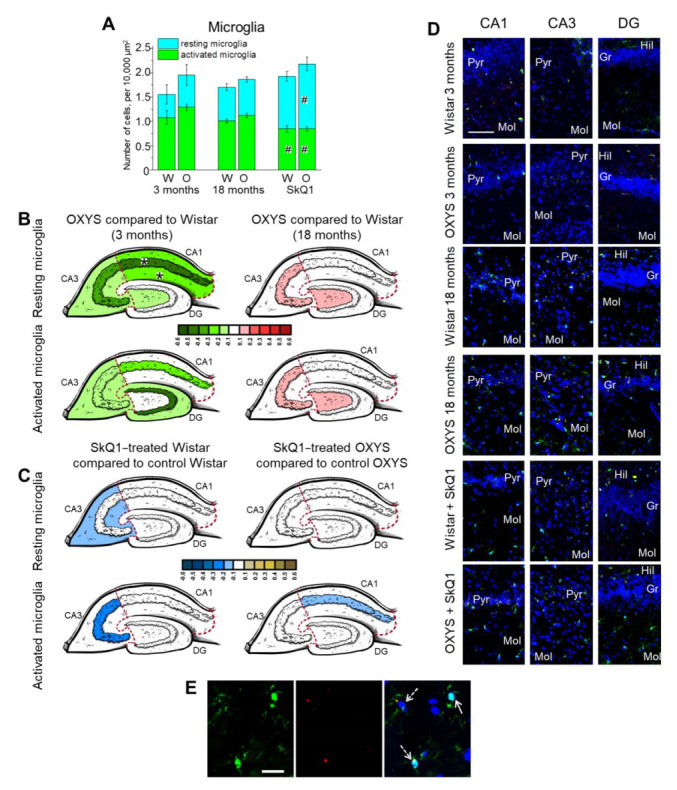
The density of resting and activated microglia in the hippocampus of OXYS and Wistar rats at 3 and 18 months of age, as well as after SkQ1 treatment (**A**). Changes in resting- and activated-microglia density in the regions of hippocampus of OXYS rats compared to Wistar rats (**B**) and in SkQ1-treated compared to untreated animals (**C**) are presented schematically. Cell density in each region and layer is coded as a heatmap on an lg scale from −0.6 to 0.6. (**D**) An image of Iba1^+^ cells (green) and CD68^+^ cells (red) in the hippocampal regions of OXYS and Wistar rats at 3 and 18 months of age, as well as after SkQ1 supplementation. The scale bar is 100 µm. (**E**) A representative image of Iba1^+^ cells (green; indicated by the arrow) and CD68^+^ cells (red; pointed out by the dashed arrow) (the molecular layer of the CA1 area in the hippocampus of a Wistar rat at 3 months of age). The scale bar is 20 µm. 4′,6-Diamidino-2-phenylindole (DAPI, blue) indicates cell nuclei (**D**,**E**). Gr: granular layer; Hil: hilus; Mol: molecular layer; O: OXYS rats; W: Wistar rats; Pyr: pyramidal layer. The data (**A**) are presented as mean ± SEM; ^#^
*p* < 0.05 for effects of SkQ1 treatment.

**Figure 5 ijms-23-01134-f005:**
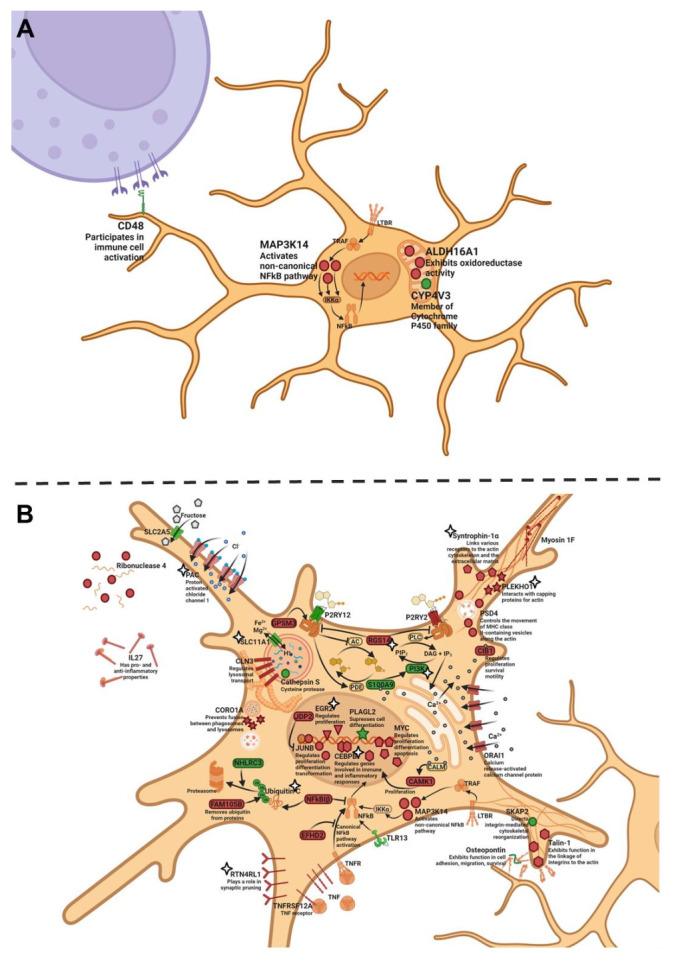
A microglial cell of 5-month-old (**A**) and 18-month-old (**B**) OXYS rats. Proteins with higher mRNA expression compared to Wistar rats or to previous age are highlighted in green; proteins with higher mRNA expression are red. Stars (**B**) indicate the genes whose expression was altered by SkQ1. The illustration was created on BioRender.com (accessed on 15 November 2021).

## Data Availability

Raw data are available from the corresponding author upon request.

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
