# Peer review of "Changes in Glial Support of the Hippocampus during the Development of an Alzheimer’s Disease-like Pathology and Their Correction by Mitochondria-Targeted Antioxidant SkQ1"

_ijms, 2022, doi:10.3390/ijms23031134_

Round 1

Reviewer 1 Report

The intended goal of this paper was to address the possible effects SkQ1 had on alterations of glial support seen in a rat model of Alzheimer's Disease.  This was done using two strains of rats at multiple ages, and examining the effects seen in the hippocampus, with a focus on different regions and cell types within the hippocampus.

General Comments:

  • The paper is in need of editing for the following:
    • Run on sentences
    • Incorrectly placed or lack of proper punctuation
    • Choice of transition words and how their connotations affect the meaning of the next sentence (ex. Line 44 - Namely, hippocampal astrocytes, Line 90 - Nonetheless, these neurodegenerative changes)
    • Missing words (ex. Line 45 - The microglial response comprises 'of' their activation)
    • Order in which the data is presented, and the best way to organize a paragraph for greatest clarity
    • Incorrect/non-existent words (ex. Line 40 - Replicatively)
  • The title of the paper does not fully represent the data presented here.  The title, as well as the abstract, indicate that the primary goal was to examine SkQ1.  However, the majority of the data presented in the paper is on the differences between OXYS and Wistar rats.
  • A lot of the results described, especially when referencing previous papers, would be better included in a review paper, rather than one presenting novel experimental results.

Abstract:

  • As with the title, the abstract focuses on SkQ1, and does not mention Wistar rats at all.
  • The abstract could use another sentence or two describing what the data mean in an overall sense, and where the next experimental steps lead.
  • The information regarding transgenic animal models (Line 13) does not give any information to the overall context.

Introduction:

  • See the previous comments regarding editing - there are a number of sentences that could be made much clearer, such as the sentence beginning on Line 39 (Moreover, ...)
  • In general, there needs to be more description/detail given.  For example, if it is mentioned that there are "persistent changes" (Line 41), it would be clarifying to include some detail regarding what these changes were, as well as any potential implications.  Another example is Line 50 - "demonstrable microglial activation".  What is meant by this?
  • Line 56-67 - while it is well known that Amyloid Beta is a marker of Alzheimer's Disease, this is the first mention of it in this paper.  All previous information was regarding glia.  This, combined with the phrasing of the second half of the sentence (after 'besides') makes the sentence unwieldy and potentially unnecessary.
  • More information is needed on why OXYS rats were used.  Alzheimer's Disease is a neurodegenerative disease, and this research examines SkQ1.  With this in mind, the fact that OXYS rats have been shown to have altered early postnatal development suggests that examining changes later in life may not be translatable to humans.  If the point is to compare Wistar and OXYS rats, the postnatal alterations would be the focus, as the goal would be to see how this early alteration resulted in later life changes.  If the goal is to examine how something can alter any Alzheimer's Disease phenotype, then starting from an altered beginning does not result in the best model with the most applicable results.
  • If the paper is intended to focus on SkQ1, then the introduction should include much more information and background on this.
  • The introduction does not mention Wistar rats, even though they comprise a good portion of the data presented.
  • Much mention is made regarding data from previous papers - see the earlier comment about potentially including things like this in a review paper, and only mentioning the results of these papers here, rather than combining the results of previous papers with the results presented here in a single sentence (Line 82, "As a result")
  • Line 73 - Examining the references suggests that this should likely be reference 24, rather than reference 23.

Results and Discussion:

  • In general, there should be a greater distinction between presenting the results, and discussion of these results.  As it is now, the writing bounces back and forth between results and discussion, rather than separating them into different portions.
  • See previous comments regarding editing, word choice, and level of detail.
  • Section 2.1:
    • The order that the results are reported should reflect the order of the figures they reference.  As it is now, ANPs are presented first, but are Figure 1B, rather than the first data presented being what is shown in Figure 1A.
    • There are places where it is unclear what comparison is being referenced, as there are comparisons between genotypes, ages and SkQ1 treatment. For example, Line 133:  ANP density was already lower at 3 months of age - density in what group, as compared to what group?
    • Figure 1A:  Is the difference between genotypes at 18 months in the treated animals no longer significant?  The data presented in this graph was mentioned to be a part of previously presented research, in a different paper.
    • When presenting results, present the results from the experiments done in this paper, and then include the results from previous papers in the paragraph(s) that discuss(es) how the results from the current experiments add to previous data.
    • Some of the information presented in the results would be better served in the introduction.
    • Most of the data presented is regarding strain differences, with comparatively little data regarding the effect of SkQ1.
  • Section 2.2
    • See previous comments regarding editing, word choice, organization of presentation, etc.
    • The first paragraph of this section is an introductory paragraph, not a results or discussion paragraph.
    • The last sentence of this first paragraph emphasizes the differences between Wistar and OXYS, rather than the influence/impact of SkQ1.
    • When presenting significant data, it needs to be clear if the significancy markers are reflecting a significant main effect or a significant interaction.
    • There are differences between Wistar and OXYS rats - see previous comments regarding more information needed as to why OXYS rats are a good fit for this model if they are beginning from an altered development.
    • Figures 2C and 2D would be more understandable in a different format.  As it is now, it is difficult to tell which group is being used as the baseline, and what group is increasing/decreasing in comparison.
    • Ensure that there is consistency between what is presented in Table S1 and what is presented in the text (ex. density of CA3 in OXYS across age)
    • The information regarding astrocyte density in the CA3 is repeated, with different statistics.
    • See previous comments regarding focus on genotypes/ages as compared to focus on SkQ1.
    • Sentence beginning on Line 244 (These data are in line with) - the hippocampus and the cortex are very different areas.  If the strains of animals examined were the same, then this comparison would be valid, but as it is the comparison is different strains as well as different areas, and is therefore an incorrect comparison to make.
    • More discussion is needed regarding why the specific areas within the hippocampus were chosen, and what it might mean that there are different changes within different areas.
  • Section 2.3
    • Figure 3 would be good for a review paper that sums up multiple sets of experiments.  As it is, it is not the best method for presenting the data in this paper.  A possible alternative would be a heat map.  Additionally, there is a difference between comparing ages and comparing strains, and it is not the best method of data presentation to group two different comparisons within a single marker (the green and red marks)
    • Why is there a switch from 3 months to 5 months?
    • Table S3 - see previous comments regarding presentation for clarity of which comparisons are being made (ex. upregulated in Wistar rats, as compared to what?  Other ages?  Other genotypes?)  Were any of the SkQ1 treated groups examined?
    • See previous comments regarding when/how to include the results from previous papers (ex. paragraph starting at Line 281 includes results from the current experiments, switches to results from previous papers, and then the subsequent paragraph details results from the current paper again).
    • When speaking about gene expression, prevented, normalized and restored all have very different meanings.  Prevented a normal increase/decrease?  Normalized to what?  Different genotypes, previous ages, lack of treatment?  Restored - does this mean that something had changed the level of expression and that treatment brought it back to "normal" levels?
  • Section 2.4
    • See previous comments regarding information that would be better included in the introduction.
    • Figure 4A - the Wistar resting microglia is not significantly different following treatment, but the activated microglia is?
    • Figure 4E - what images are these?  What animals?  Groups?  Treatments?  Brain regions?  Bregma?  Where is the mentioned scale bar?
    • See previous comments regarding needing a clearer way of indicating an interaction, as opposed to a single main effect.
    • The text (paragraph starting on Line 400) mention comparisons between microglia and astrocytes that were done for these experiments.  Where are the data/figures for this?
    • See previous comments regarding presenting the novel data of the experiments described in and for this specific paper and the need to separate it from paragraphs presenting data from previous papers, as well as the comments regarding the potential for a review paper.
    • Line 440 "Thus, SkQ1 treatment.....rats from both strains" - Table S6 shows that only Wistar rats had a significant difference.
  • Section 2.5
    • See previous comments regarding information that would be better put in an introduction.
    • See previous comments regarding 5 months of age animals.
    • See previous comments regarding editing and missing words.
    • The statistics here (FDR) are different from the previously described gene analysis' statistics.
    • Figure 5 - see comments about Figure 3 for applicability for a review paper vs. experimental results.  Clarification/simplification necessary.
    • See previous comments regarding comparisons of genotype and age vs. comparisons via SkQ1.

Materials and Methods:

  • See previous comments regarding editing.
  • An explanation is needed for why only male animals were used.  Alzheimer's Disease impacts both sexes in humans, and it is not scientifically sound to assume that results seen in one sex, even in an animal model, are directly and completely applicable to the other sex.  This is doubly true when a treatment or drug is being examined, as there have been previous instances where a sex difference in reaction to the drug/dose of drug has resulted in health complications.
  • SkQ1 administration is described as "oral" (Line 514).  Clarification is needed - intubation?  Included in the food?
  • Why were dried bread slices used?
  • See previous comments regarding use of both OXYS and Wistar rats.
  • The cell density calculations (paragraph starting on Line 553) needs a great deal more explanation.  Why was this unique method of density calculation used?  Have other studies used this method?  What are the advantages that using this method presents?

Conclusions:

  • A great deal more detail and explanation are necessary.
  • This paragraph is not really a conclusion of the data presented in this paper, but is instead an overall generalization of the results compiled with the results from previous papers, with no discussion regarding implications or future avenues of research.

Author Response

The intended goal of this paper was to address the possible effects SkQ1 had on alterations of glial support seen in a rat model of Alzheimer's Disease.  This was done using two strains of rats at multiple ages, and examining the effects seen in the hippocampus, with a focus on different regions and cell types within the hippocampus.

General Comments:

The paper is in need of editing for the following:

  1. Run on sentences
  2. Incorrectly placed or lack of proper punctuation
  3. Choice of transition words and how their connotations affect the meaning of the next sentence (ex. Line 44 - Namely, hippocampal astrocytes, Line 90 - Nonetheless, these neurodegenerative changes)
  4. Missing words (ex. Line 45 - The microglial response comprises 'of' their activation)
  5. Order in which the data is presented, and the best way to organize a paragraph for greatest clarity
  6. Incorrect/non-existent words (ex. Line 40 - Replicatively)

Thank you for noticing; we implemented all these suggestions. We had the manuscript checked by a professional language-editing company.

The title of the paper does not fully represent the data presented here.  The title, as well as the abstract, indicate that the primary goal was to examine SkQ1.  However, the majority of the data presented in the paper is on the differences between OXYS and Wistar rats.

The Wistar strain is a normal (healthy) control (parental strain) and therefore helped us to determine whether the effects of SkQ1 were beneficial. In the present study, we investigated the changes of hippocampal neurogenesis and glial support of the hippocampus during the development and progression of an Alzheimer’s disease-like pathology. In addition, here we continued to analyze the effect of prolonged supplementation (from age 12 to 18 months) with mitochondria-targeted antioxidant SkQ1 on the neurogenesis and glial support of the hippocampus.

A lot of the results described, especially when referencing previous papers, would be better included in a review paper, rather than one presenting novel experimental results.

Abstract:

  1. As with the title, the abstract focuses on SkQ1, and does not mention Wistar rats at all.

We now mention Wistar rats as a parental normal strain in the abstract.

  1. The abstract could use another sentence or two describing what the data mean in an overall sense, and where the next experimental steps lead.

As requested, we tried to update the abstract; unfortunately, we are limited to 200 words by the journal’s requirements.

  1. The information regarding transgenic animal models (Line 13) does not give any information to the overall context.

We revised the sentence to clarify it and removed the mention of transgenic animals.

Introduction:

  1. See the previous comments regarding editing - there are a number of sentences that could be made much clearer, such as the sentence beginning on Line 39 (Moreover, ...)

Sorry for this oversight. We revised the sentence for clarity.

  1. In general, there needs to be more description/detail given.  For example, if it is mentioned that there are "persistent changes" (Line 41), it would be clarifying to include some detail regarding what these changes were, as well as any potential implications.  Another example is Line 50 - "demonstrable microglial activation".  What is meant by this?

Thanks for the comments. We added information related to changes in senescent astrocytes; as for microglial activation, the phenomenon is described above in this paragraph, and we revised the sentence to make it clearer.

  1. Line 56-67 - while it is well known that Amyloid Beta is a marker of Alzheimer's Disease, this is the first mention of it in this paper.  All previous information was regarding glia.  This, combined with the phrasing of the second half of the sentence (after 'besides') makes the sentence unwieldy and potentially unnecessary.

Sorry about this problem. We tried to clarify the meaning of this sentence.

  1. More information is needed on why OXYS rats were used.  Alzheimer's Disease is a neurodegenerative disease, and this research examines SkQ1.  With this in mind, the fact that OXYS rats have been shown to have altered early postnatal development suggests that examining changes later in life may not be translatable to humans.  If the point is to compare Wistar and OXYS rats, the postnatal alterations would be the focus, as the goal would be to see how this early alteration resulted in later life changes.  If the goal is to examine how something can alter any Alzheimer's Disease phenotype, then starting from an altered beginning does not result in the best model with the most applicable results.

We agree; we clarified the goal of the study.

  1. If the paper is intended to focus on SkQ1, then the introduction should include much more information and background on this.

We appreciate the suggestion. SkQ1 is important but not the major focus of the manuscript. We corrected the part of the Introduction section dealing with SkQ1.

  1. The introduction does not mention Wistar rats, even though they comprise a good portion of the data presented.

We now mention Wistar rats in the Introduction as a parental normal (healthy) rat strain (control).

  1. Much mention is made regarding data from previous papers - see the earlier comment about potentially including things like this in a review paper, and only mentioning the results of these papers here, rather than combining the results of previous papers with the results presented here in a single sentence (Line 82, "As a result")

We updated the text accordingly (lines 116-119).

  1. Line 73 - Examining the references suggests that this should likely be reference 24, rather than reference 23.

Thank you for the perusal of our paper. Sorry to disagree, but the references are presented in the correct order.

Results and Discussion:

In general, there should be a greater distinction between presenting the results, and discussion of these results.  As it is now, the writing bounces back and forth between results and discussion, rather than separating them into different portions.

See previous comments regarding editing, word choice, and level of detail.

We moved some “discussion” parts either to the Introduction or to paragraphs below within a respective subsection or deleted them.

Section 2.1:

  1. The order that the results are reported should reflect the order of the figures they reference.  As it is now, ANPs are presented first, but are Figure 1B, rather than the first data presented being what is shown in Figure 1A.

As suggested, we arranged the parts of the figure in the appropriate order.

  1. There are places where it is unclear what comparison is being referenced, as there are comparisons between genotypes, ages and SkQ1 treatment. For example, Line 133:  ANP density was already lower at 3 months of age - density in what group, as compared to what group?

We updated comparisons throughout the Results and Discussion section.

  1. Figure 1A:  Is the difference between genotypes at 18 months in the treated animals no longer significant?  The data presented in this graph was mentioned to be a part of previously presented research, in a different paper.

Thank you for the comment; however, it should be mentioned that all data presented in graphs across the manuscript haven’t been published before. We updated the sentence in question (lines 158–159). Besides, we did not compare the SkQ1-treated OXYS group with the SkQ1-treated Wistar group. We were interested in effects of SkQ1 in comparison with untreated animals of each strain.

  1. When presenting results, present the results from the experiments done in this paper, and then include the results from previous papers in the paragraph(s) that discuss(es) how the results from the current experiments add to previous data.

We rewrote the results keeping the comment in mind.

  1. Some of the information presented in the results would be better served in the introduction.

We moved some information from Results and Discussion to the Introduction section.

  1. Most of the data presented is regarding strain differences, with comparatively little data regarding the effect of SkQ1.

Unfortunately, we did not observe a significant effect of SkQ1 treatment on QNP and ANP densities in OXYS and Wistar rats (lines 154–156).

Section 2.2

  1. See previous comments regarding editing, word choice, organization of presentation, etc.

We revised the text accordingly.

  1. The first paragraph of this section is an introductory paragraph, not a results or discussion paragraph.

We agree. The text was moved to Introduction.

  1. The last sentence of this first paragraph emphasizes the differences between Wistar and OXYS, rather than the influence/impact of SkQ1.

We added a phrase about SkQ1 effects.

  1. When presenting significant data, it needs to be clear if the significancy markers are reflecting a significant main effect or a significant interaction.

Sorry for this oversight; we now mention the types of effects throughout the results.

  1. There are differences between Wistar and OXYS rats - see previous comments regarding more information needed as to why OXYS rats are a good fit for this model if they are beginning from an altered development.

You are making a good point; this model is not perfect but reproduces many pathological changes seen in Alzheimer’s disease; we believe it’s a good model as compared to other available models. We have shown in many previous publications that signs of AD spontaneously develop in OXYS rats. Besides, a sequence of these events in OXYS rats is in line with the modern AD hypotheses, which point to amyloid accumulation as a consequence rather than a cause of sporadic AD [Drachman, 2014; Mecocci et al., 2018; Fan et al., 2020]. To reveal the key events leading to the AD-like pathology in OXYS rats, we started to investigate brain development as the most vulnerable period of brain function. Surprisingly to us, but in line with current AD research [Schaefers, Teuchert-Noodt, 2013; Faa et al., 2014; Fanni et al., 2016], we found some alterations of brain development in OXYS rats when compared to the parental normal Wistar rat strain. It is important to point out that by the end of adolescence, all interstrain differences disappear.

  1. Figures 2C and 2D would be more understandable in a different format.  As it is now, it is difficult to tell which group is being used as the baseline, and what group is increasing/decreasing in comparison.

We tried to present the data in Figures 2C and 2D more clearly.

  1. Ensure that there is consistency between what is presented in Table S1 and what is presented in the text (ex. density of CA3 in OXYS across age)

As suggested, we checked all the data for consistency.

  1. The information regarding astrocyte density in the CA3 is repeated, with different statistics.

Thanks for noticing, we removed the repeated data.

  1. See previous comments regarding focus on genotypes/ages as compared to focus on SkQ1.

We revised the text as much as we could.

  1. Sentence beginning on Line 244 (These data are in line with) - the hippocampus and the cortex are very different areas.  If the strains of animals examined were the same, then this comparison would be valid, but as it is the comparison is different strains as well as different areas, and is therefore an incorrect comparison to make.

This is a valid point. We rephrased the sentence to avoid such a comparison (lines 278-282).

  1. More discussion is needed regarding why the specific areas within the hippocampus were chosen, and what it might mean that there are different changes within different areas.

We added the clarification concerning the chosen areas into the Introduction section (lines 108–112).

Section 2.3

  1. Figure 3 would be good for a review paper that sums up multiple sets of experiments.  As it is, it is not the best method for presenting the data in this paper.  A possible alternative would be a heat map.  Additionally, there is a difference between comparing ages and comparing strains, and it is not the best method of data presentation to group two different comparisons within a single marker (the green and red marks)

We appreciate the suggestion. Sorry to disagree, but here we are using the combined format (one Results and Discussion section); therefore, the current form of presentation of the results seems to be appropriate as a summarizing scheme; the manuscript is already rather long. The RNA-Seq data separated by time and strain are presented in tables in the Supplementary materials.

  1. Why is there a switch from 3 months to 5 months?

During the period of 3–5 months of age, OXYS rats are within the same stage of the development of AD-like pathology. Thus, it seems appropriate to use animals from this age group to study the manifestation stage of AD signs.

  1. Table S3 - see previous comments regarding presentation for clarity of which comparisons are being made (ex. upregulated in Wistar rats, as compared to what?  Other ages?  Other genotypes?)  Were any of the SkQ1 treated groups examined?

Thank you for the remark; we revised column titles in Table S3 to make it clearer.

  1. See previous comments regarding when/how to include the results from previous papers (ex. paragraph starting at Line 281 includes results from the current experiments, switches to results from previous papers, and then the subsequent paragraph details results from the current paper again).

Thank you for the meticulous review. The paragraph in question contains results from the new experiment; only in the last sentence of this paragraph do we refer to our previous study in the context of the discussion of obtained results. Please note that this is a Results and Discussion section.

  1. When speaking about gene expression, prevented, normalized and restored all have very different meanings.  Prevented a normal increase/decrease?  Normalized to what?  Different genotypes, previous ages, lack of treatment?  Restored - does this mean that something had changed the level of expression and that treatment brought it back to "normal" levels?

As requested, we clarified the presentation of the results. We meant that variables were moved closer to those seen in the normal control strain: Wistar rats.

Section 2.4

  1. See previous comments regarding information that would be better included in the introduction.

We revised the text accordingly.

  1. Figure 4A - the Wistar resting microglia is not significantly different following treatment, but the activated microglia is?

Indeed, in Wistar rats, the density of resting microglia was not affected by SkQ1 treatment, but the density of activated microglia decreased after SkQ1 treatment.

  1. Figure 4E - what images are these?  What animals?  Groups?  Treatments?  Brain regions?  Bregma?  Where is the mentioned scale bar?

Thank you very much for your attention to detail; we inserted all the missing information.

  1. See previous comments regarding needing a clearer way of indicating an interaction, as opposed to a single main effect.

We now describe interactions better and specify single main effects.

  1. The text (paragraph starting on Line 400) mention comparisons between microglia and astrocytes that were done for these experiments.  Where are the data/figures for this?

Sorry and thank you noticing, we added Figure S1 to Supplementary materials.

  1. See previous comments regarding presenting the novel data of the experiments described in and for this specific paper and the need to separate it from paragraphs presenting data from previous papers, as well as the comments regarding the potential for a review paper.

As suggested, we carefully revised the text.

  1. Line 440 "Thus, SkQ1 treatment.....rats from both strains" - Table S6 shows that only Wistar rats had a significant difference.

We agree; we rewrote the sentence to make it clearer.

Section 2.5

  1. See previous comments regarding information that would be better put in an introduction.

We implemented this suggestion here.

  1. See previous comments regarding 5 months of age animals.

During the period of 3–5 months of age, OXYS rats are within the same stage of the development of AD-like pathology. Therefore, it appears to be appropriate to use the rats from this age group to study the manifestation stage of AD signs.

  1. See previous comments regarding editing and missing words.

We had the manuscript checked by a professional company. Thank you for the remark; we clarified the data in the Table S7.

  1. The statistics here (FDR) are different from the previously described gene analysis' statistics.

We apologize for this inconsistency, we corrected it.

  1. Figure 5 - see comments about Figure 3 for applicability for a review paper vs. experimental results.  Clarification/simplification necessary.

We appreciate this suggestion. Sorry to disagree, but here we are using the combined format (one Results and Discussion section); therefore, the current form of presentation of the results seems to be appropriate as a summarizing scheme; the manuscript is already rather long. The RNA-Seq data stratified by age and strain are given in tables in the Supplementary materials.

  1. See previous comments regarding comparisons of genotype and age vs. comparisons via SkQ1.

We updated the text for clarity.

Materials and Methods:

  1. See previous comments regarding editing.
  2. An explanation is needed for why only male animals were used.  Alzheimer's Disease impacts both sexes in humans, and it is not scientifically sound to assume that results seen in one sex, even in an animal model, are directly and completely applicable to the other sex.  This is doubly true when a treatment or drug is being examined, as there have been previous instances where a sex difference in reaction to the drug/dose of drug has resulted in health complications.

This is an excellent observation. We work with male rats to avoid the effects of the hormonal cycle characteristic of female rats (the clarification is now provided, lines 623–624). However, it should be mentioned that the current work is not a preclinical trial of the drug. We tried to shed light on some aspects of AD development using the model of sporadic AD and to evaluate the ability of SkQ1 to influence these AD parameters. Without a doubt, further studies are needed, including studies on female rats.

  1. SkQ1 administration is described as "oral" (Line 514).  Clarification is needed - intubation?  Included in the food?

Sorry about the lack of clarity. We updated the text (lines 627-630). We added a solution of SkQ1 onto a dried bread slice; the amount of SkQ1 was normalized to rat body weight. Then each rat in the cage received its own bread slice and ate it whole.

  1. Why were dried bread slices used?

The animals received dried bread slices (one bread slice per day, individually) in addition to standard rodent feed before the study was started. The rats were accustomed to consuming dried bread slices and liked them very much. Thus, adding the SkQ1 solution to a bread slice during an experiment is a simple and nonstressful way to administer an exact amount of SkQ1 per os (clarification added: lines 633-634).

  1. See previous comments regarding use of both OXYS and Wistar rats.

We added information about the OXYS strain origin in the Materials and Methods section.

  1. The cell density calculations (paragraph starting on Line 553) needs a great deal more explanation.  Why was this unique method of density calculation used?  Have other studies used this method?  What are the advantages that using this method presents?

We used a standard histological approach [von Bartheld et al., 2016] for counting the total number of cell nuclei (stained by DAPI) colocalized with processes stained by antibodies to specific markers of each cell type [Encinas et al., 2011; González Ibanez et al., 2019]. The only possible deviation from the standard method may be the choice of an area of interest. We did not count cells in several fields of view; instead, we determined the cell number in the whole hippocampus as well as measured the area of the whole hippocampus in images. Then, we normalized the total cell number to the total hippocampal area.

Conclusions:

  1. A great deal more detail and explanation are necessary.

We now provide detailed clarifications throughout the Results and Discussion section and more lucid conclusions in the last paragraph of this section. We hope this is acceptable.

  1. This paragraph is not really a conclusion of the data presented in this paper, but is instead an overall generalization of the results compiled with the results from previous papers, with no discussion regarding implications or future avenues of research.

We removed the Conclusion section and now provide concluding remarks after each subsection of Results and Discussion instead.

Reviewer 2 Report

The authors have taken a multifaceted approach to assessing the changes in neurogenesis glial support in a sporadic AD model, OXYS rats. The research team has previously published this model extensively. As the authors point out, these studies are crucial for understanding the majority of AD cases, which are sporadic. Importantly, they have included to time points: 3 and 18 mo to study the aging component- crucial for AD studies. It is a nice, well-written and comprehensive manuscript with well thought-out experiments, proper statistical analysis and clear experimental goals.  

Introduction:

  • …“also persistent changes in gene expression and epigenetic regulation, alterations in mitochondrial homeostasis, and disruption of energy and cellular metabolism”… a verb is missing here (occur?).
  • The origin of OXYS rats should be explained in the intro.

Results:

Words should be written out in full form before abbreviations are used in figure legends (e.g. DG, QNP, ANP)

  • Fig 1C should include arrows depicting what was quantified as an ANP and QNP.
  • 5, lines 158-192- results should include reference to figure 2 data

Material and methods

  • 1: the housing conditions (single or group and number) should be included; authors should mention what OXYS stands for.
  • 2: please clarify why bread slices were given to the rats- were they impregnated with SkQ1? And if so, how?
  • 3: “To determine the percentage of dying cells consumed by microglia, we counted all pyknotic nuclei and the pyknotic nuclei covered by microglial cytoplasm”. The authors should consider adding a representative image of a measured pyknotic nucleus in the SI.

Author Response

The authors have taken a multifaceted approach to assessing the changes in neurogenesis glial support in a sporadic AD model, OXYS rats. The research team has previously published this model extensively. As the authors point out, these studies are crucial for understanding the majority of AD cases, which are sporadic. Importantly, they have included to time points: 3 and 18 mo to study the aging component- crucial for AD studies. It is a nice, well-written and comprehensive manuscript with well thought-out experiments, proper statistical analysis and clear experimental goals.  

Thank you very much for the positive evaluation of our study.

Introduction:

  1. …“also persistent changes in gene expression and epigenetic regulation, alterations in mitochondrial homeostasis, and disruption of energy and cellular metabolism”… a verb is missing here (occur?).

Thanks for noticing, we split and revised this sentence (lines 52–54).

  1. The origin of OXYS rats should be explained in the intro.

We added this information into the Introduction and Materials and Methods sections.

Results:

Words should be written out in full form before abbreviations are used in figure legends (e.g. DG, QNP, ANP)

As requested, we redefined abbreviations in figure legends. Please be advised that protein symbols are not considered abbreviations.

  1. Fig 1C should include arrows depicting what was quantified as an ANP and QNP.

We agree; we added a zoomed-in part of Figure 1C to visualize the ANPs and QNPs.

  1. 5, lines 158-192- results should include reference to figure 2 data

 We inserted additional references to Figure 2 data.

Material and methods

  1. the housing conditions (single or group and number) should be included; authors should mention what OXYS stands for.

We added information about housing conditions into the Materials and Methods section. Sorry, OXYS is not an abbreviation; it’s a strain name of obscure origin.

  1. please clarify why bread slices were given to the rats- were they impregnated with SkQ1? And if so, how?

We added a solution of SkQ1 onto a dried bread slice; the amount of SkQ1 was normalized to rat body weight. Then, each rat in the cage received its own bread slice and ate it whole (lines 627–632).

  1. “To determine the percentage of dying cells consumed by microglia, we counted all pyknotic nuclei and the pyknotic nuclei covered by microglial cytoplasm”. The authors should consider adding a representative image of a measured pyknotic nucleus in the SI.

Thank you for the comment; we now provide a representative image of a pyknotic nucleus in the Supplementary materials.

Round 2

Reviewer 1 Report

The authors are to be commended for their thorough approach to correcting the proposed revisions and answering any questions.  The paper should be accepted as is, but I would like to reiterate that both sexes are needed for accurate scientific investigation.  I would like to highly encourage the authors to most definitely include both sexes in all future research, regardless of any preconceptions regarding variability in females.

Author Response

The authors are to be commended for their thorough approach to correcting the proposed revisions and answering any questions.  The paper should be accepted as is, but I would like to reiterate that both sexes are needed for accurate scientific investigation.  I would like to highly encourage the authors to most definitely include both sexes in all future research, regardless of any preconceptions regarding variability in females.

Thank you very much for the comment; we are taking it into account and including both sexes in the planned experiment.